# Third delay in care of critically ill patients: a qualitative investigation of public hospitals in Kenya

Onesmus O Onyango ![ORCID],[1] Tamara M Willows,[2] Jacob McKnight,[2] Carl Otto Schell ![ORCID],[3,4,5] Tim Baker ![ORCID],[3,6,7,8] Elibariki Mkumbo,[6] John Maiba,[6] Karima Khalid,[6,9] Mike English ![ORCID],[1,2] Jacquie N Oliwa ![ORCID] [1]

For numbered affiliations see end of article.

**Correspondence to**
Onesmus O Onyango;
oonyango@kemri-wellcome.org

## ABSTRACT

**Objectives** Third delay refers to delays in delivering requisite care to patients after they arrive at a health facility. In low-resource care settings, effective triage and flow of care are difficult to guarantee. In this study, we aimed to identify delays in the delivery of care to critically ill patients and possible ways to address these delays.

**Design** This was an exploratory qualitative study using in-depth interviews and patient journeys. The qualitative data were transcribed and aggregated into themes in NVivo V.12 Plus using inductive and deductive approaches.

**Setting** This study was conducted in four secondary-level public Kenyan hospitals across four counties between March and December 2021. The selected hospitals were part of the Clinical Information Network.

**Participants** Purposive sampling method was used to identify administrative and front-line healthcare providers and patients. We conducted 12 in-depth interviews with 11 healthcare workers and patient journeys of 7 patients. Informed consent was sought from the participants and maintained throughout the study.

**Results** We identified a cycle of suboptimal systems for care with adaptive mechanisms that prevent quality care to critically ill patients. We identified suboptimal systems for identification of critical illness, inadequate resources for continuity care and disruption of the flow of care, as the major causes of delays in identification and the initiation of essential care to critically ill patients. Our study also illuminated the contribution of inflexible bureaucratic non-clinical business-related organisational processes to third delay.

**Conclusion** Eliminating or reducing delays after patients arrive at the hospital is a time-sensitive measure that could improve the care outcomes of critically ill patients. This is achievable through an essential emergency and critical care package within the hospitals. Our findings can help emphasise the need for standardised effective and reliable care priorities to maintain of care of critically ill patients.

## STRENGTHS AND LIMITATIONS OF THIS STUDY

⇒ This study draws strength from the multiple methods employed, which with triangulation, yielded valuable evidence and support.
⇒ The study was conducted in facilities operating at the same level from four counties across different regions of the country.
⇒ The agreement of various staff from the hospitals with our findings as being a true reflection of their various facilities provided a validation for our study findings.
⇒ However, a wider sample of facilities, healthcare providers, and patients would likely provide more robust evidence to explore our aim.
⇒ Although the use of video for interviewing during COVID-19 has been justified, it is suboptimal relative to in-person interviews.

and treating patients with critical illness through the initial and sustained support of vital organ functions. Critically ill patients, therefore, require prompt, effective care wherever they are in health facilities to reduce the risk of dying. Although the true burden is unknown, the morbidity and mortality from critical illness are thought to be very high and constitute a considerable contribution to the global burden of disease.[2–6] Many of the deaths caused by critical illness are potentially preventable[7 8] if a prompt, basic, essential package of care (online supplemental appendix 1) is provided to these patients.[9]

Critical illness gained significant global attention during the COVID-19 pandemic but strategies to improve the management of critically ill patients, especially in resource-constrained settings are still in their infancy. With the overwhelming cases of critically ill patients during the COVID-19 pandemic, the Kenyan government—through the national and respective county governments—embarked on setting up COVID-19 isolation units and intensive care units (ICUs) as well

## BACKGROUND

Critical illness is defined as a 'state of ill health with vital organ dysfunction, a high risk of imminent death if care is not provided and the potential for reversibility'.[1] Critical care is here described as identifying, monitoring

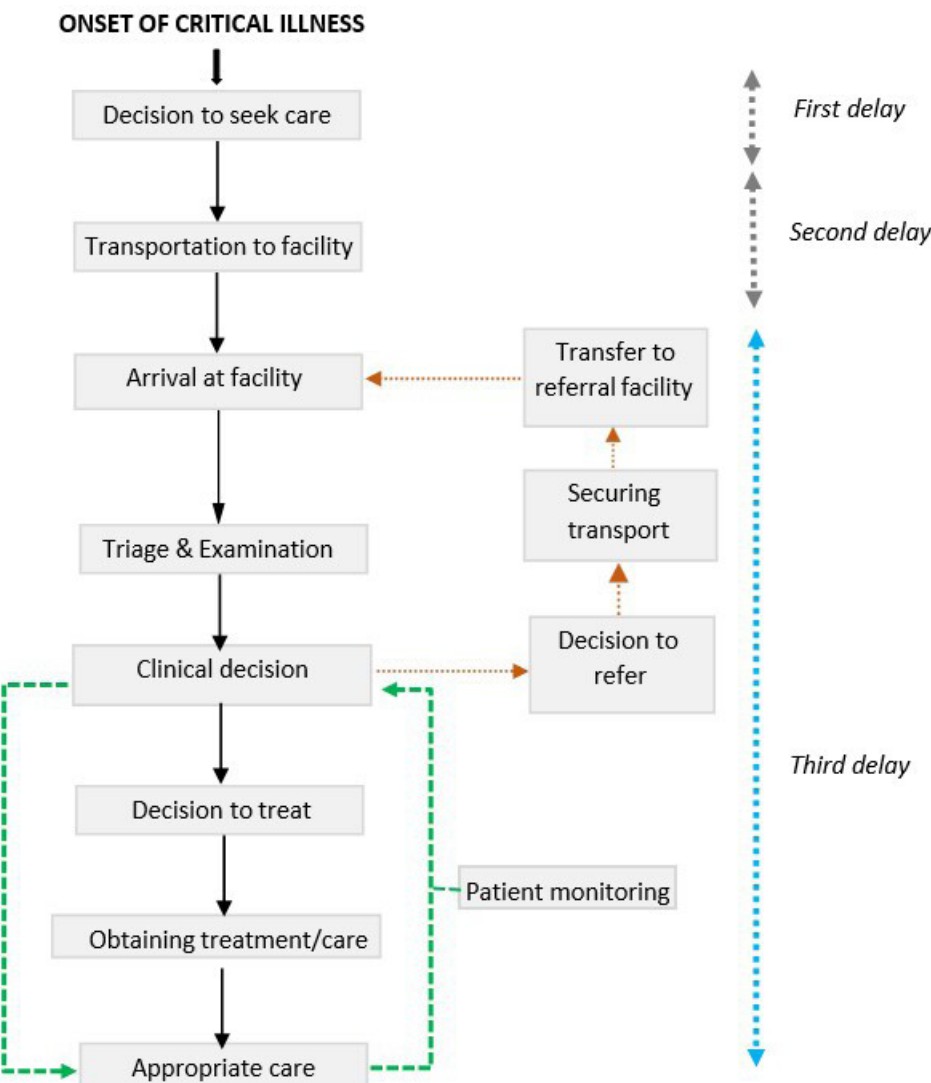

**Figure 1** The framework of the third delay model (adapted from Gabrysch and Campbell, and Cavallaro and Marchant).[18] The loop represented by the green connectors illustrates the continuous reassessment and monitoring of acutely ill patients needed to identify critical illness. The loop represented by the orange dotted connectors represents a second referral to a higher-level facility. Any complications that cannot be managed at the first facility should be promptly diagnosed and referred within the shortest time possible to eliminate second emergency delay.

as training healthcare workers to improve intensive care capacity.[10] However, these efforts were limited by weak public hospital systems and marked health workforce deficits,[11 12] and facilities continued to see an upsurge of critically ill patients.

Triage is the basis of early identification of critical illness and consequently the first step in the management and survival of critically ill patients in health facilities that attend to large volumes of patients.[13–16] A critically ill patient should ideally be examined immediately on arrival, leading to a decision on the priority level of the patient's condition and their management. For appropriate triage and care delivery, hospitals must have basic treatment materials, skilled personnel with full awareness of the resources and, if necessary, means of transportation for treatment or referrals to be carried out.[17–19] Available evidence suggests that triage, basic emergency care

training, better flow in hospitals and supervision of junior emergency providers reduces care delays and mortality in low-resource settings.[20] In low-resource settings, effective triage and flow of care are, however, difficult to accomplish. Healthcare facilities are frequently characterised by overcrowding, insufficient drugs, limited equipment and staff shortages[13 21 22] as well us a lack of understanding of critical illness,[23] contributing directly to delay in essential care. To use the limited human and material resources effectively, different forms of triage are used to classify patients according to the severity of their condition and determine the priority for further treatment.[24 25]

The three delays model[26 27] (figure 1) has been used for decades in obstetrics to illuminate obstacles to the provision of timely and quality care. The first delay refers to delays in making the decision to seek medical care, mostly due to family and community reasons. Second

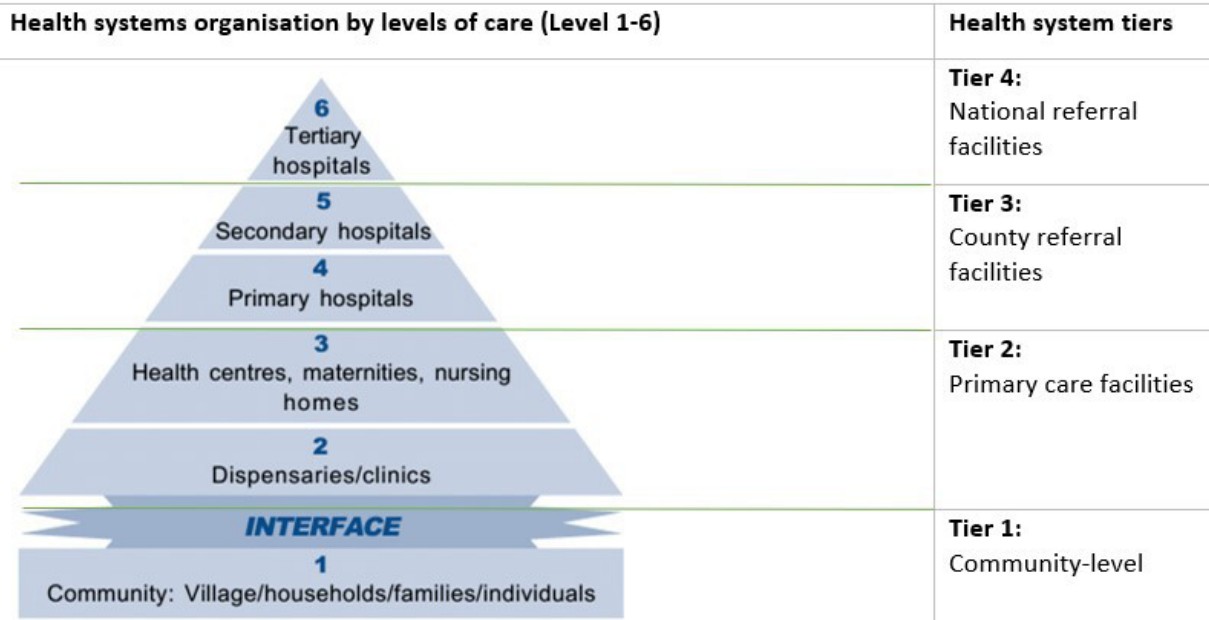

**Figure 2** Organisation of the Kenyan public health system.

delay refers to delays in arriving at a health facility after the decision to seek care has been made due to accessibility challenges. The third delay is concerned with delay in delivering care to patients after arrival to a health facility. Reducing all delays improves the clinical outcome of patients both in and outside the hospital, but reducing the third delay directly reduces in-hospital morbidity and mortality by adopting a holistic approach to understand the responsibilities of hospital systems.

While the third delay model helps us understand and address care gaps in maternal health in resource-constrained settings, the methods and findings of this approach have not been applied substantively to the delivery of care to critically ill patients, especially in low-resource hospitals, outside the obstetric domain. Application of the third delay concept through simple strategies such as mapping workflows could, therefore, be a useful model to identify points of urgent intervention for all patients presenting with critical illness in these settings. Therefore, using this concept, we aimed to identify delays in the delivery of care to critically ill patients in Kenyan secondary-level public hospitals.

## METHODS
### Study design
This was an exploratory qualitative study design to observe care provided to critically ill patients. The study was part of a larger multimethod project—Provision of Essential Treatment in Critical Illness in the COVID-19 pandemic (POETIC-COVID)—that sought to investigate EECC[9] in Kenyan hospitals during the pandemic. We used in-depth interviews (IDIs) and patient journeys across the selected secondary referral-level hospitals and analysed the results in detail using inductive and deductive approaches.

### Study setting
Kenya, a lower-middle-income country, has a population of approximately 48 million people served by 907 hospitals across 47 counties operating at primary, secondary and tertiary levels[28] (figure 2). The study was conducted in four secondary-level public Kenyan hospitals (figure 2) across four of the 47 counties in Kenya. The hospitals selected for our study were part of the Clinical Information Network[29] and were selected purposively[30] in four counties based on the level of care they provide and their accessibility from Nairobi, where the lead researcher was based, owing to COVID-19 restrictions. All four hospitals included in our study had received funding to scale up critical illness care provision during the COVID-19 pandemic. These hospitals had bed capacities ranging from 185 to 449.

Table 1 outlines the staffing structure across the departments in the four hospitals. Low staffing numbers were picked across departments in the four facilities. These data were collected in the health facility assessment (HFA) of these hospitals that was a preliminary stage of the POETIC-COVID study.

### Participant selection and recruitment
A purposive sampling method[30] was used to identify administrative staff and front-line healthcare providers from the emergency and outpatient departments, ICUs, maternity units and general wards (paediatric and adult wards). Using staff clinical judgement, the patients were picked from the emergency departments that typically received critically ill patients in these hospitals.

We explained to the eligible participants the purpose and procedure of the study, and that privacy and confidentiality, and their right to opt out at

**Table 1** Staffing structure across the facilities during the visit (numbers represent one shift)

| Department | Staff cadre | Staff-to-patient ratios (day shift) | | | |
|---|---|---|---|---|---|
| | | Facility 1 | Facility 2* | Facility 3 | Facility 4 |
| Accident and Emergency (A&E) | N | 4 | 5 | 3 | 2 |
| | CO | 1 | 4 | 0 | 2 |
| | MO† | 1 | 2 | 1 | 1 |
| Obstetrics | N | 10:12 | 8:37 | 9:73 | 2:19 |
| | CO | 2:12 | 2:37 | 0 | 0 |
| | MO† | 2:12 | 5:37 | 2:73 | 1:19 |
| Paediatric ward | N | 4:48 | 4:25 | 3:34 | 1:22 |
| | CO | 5:48 | 4:25 | 2:34 | 1:22 |
| | MO† | 2:48 | 3:25 | 1:34 | 2:22 |
| Medical wards (male+female) | N | 8:45 | 4:6 | 8:35 | 2:15 |
| | CO | 4:45 | 4:6 | 0 | 0 |
| | MO† | 2:45 | 4:6 | 2:35 | 1:15 |
| Surgical wards (male+female) | N | 8:50 | 3:8 | 8:51 | 3:12 |
| | CO | 2:50 | 3:8 | 0:51 | 0:12 |
| | MO† | 1:50 | 4:8 | 1:51 | 1:12 |
| ICU | N | 3:1 | – | 9:5 | – |
| | CO | 1:1 | | 1:5 | |
| | MO | 1:1 | | 0 | |

The patient numbers in the A&E department were not included since this was fluctuant across the observation shift.
*Multiple departments reduced their patient intake due to refurbishment work in facility. previous bed capacity for the medical and surgical departments was 32 beds each.
†The MOs across all facilities covered more than one department during their shift.
CO, clinical officer; ICU, intensive care unit; MO, medical officer; N, nurse.

their convenience would be upheld. The number of participants to interview was guided by thematic saturation,[31 32] that is, when no new findings were emerging. Individual written consent was sought from the healthcare workers regarding participation and recording of the interviews and observations. We obtained verbal consent from the patients and/or their relatives about observing the care they received along the pathways.

It was explained that anonymity would be ensured regarding the use of the transcribed interviews, observations and voice notes in the patient journeys. We explained to the participants that all the audio recordings pertaining to the walkthroughs would be deleted once the recordings were transcribed. Only participants who gave informed consent were recruited in this study.

## Data collection
### In-depth interviews
For the four study hospitals, we used a semistructured interview guide (online supplemental appendix 2) to conduct a total of 12 IDIs with 11 front-line healthcare workers—nurses, clinical officers, medical officers—and administrative staff. These interviews were conducted virtually via Microsoft Teams and phone calls owing to the COVID-19 restrictions regarding face-to-face meetings. The interviews, averaging 43 min, were audiorecorded and saved onto password-protected cloud storage database managed by the Kenya Medical Research Institute-Wellcome Trust Research Programme (KWTRP). All audiorecordings were transcribed verbatim and were deleted after transcription.

### Process mapping and patient journeys
Patient journey mapping is a deliberate patient-oriented approach to better understand barriers, facilitators, experiences, interactions with services and/or outcomes for individuals and/or their caregivers and family members as they enter, navigate, experience and exit service points in a health system by documenting elements of the journey to produce a visual or descriptive map.[33 34]

The patient journey approach has been applied in patient safety to analyse the care service system and support human-centred design.[35 36] The approach focuses on the sequence of care from the patient's perspective and allows for the examination of discrete patient interactions with providers at each encounter.[35] This can be used to identify gaps that might otherwise go unnoticed.

The observer—the lead author OOO—a research assistant with background training in nursing, conducted the

journey mapping during a HFA of the four hospitals for the larger POETIC-COVID project. He first identified eight front-line healthcare workers (two per hospital) working in the emergency departments and through informal conversations using scenario-based questions, mapped the journey of a hypothetical critically ill patient when they arrived in the hospitals from the healthcare workers' perspective. OOO drew the process maps as described by the health workers in illustrative flow diagrams (online supplemental appendix 3). The healthcare workers were asked to estimate the time taken at each step along the journey.

With the help of the staff at the emergency and outpatient departments, the observer then purposively selected and followed seven critically ill patients as determined by their signs and symptoms from the time of arrival at the outpatient/emergency department to the point of admission into the wards or referral out to another facility. The observer noted the care provided and periods of inactivity against the time. Using a Dictaphone, the observer made real-time onsite audio commentary of the steps and activities against the time performed and this data was used to write out each patient journey.[35 37]

To explore third delay issues, we assessed the time taken in identification and administration of appropriate care for the selected patients. We used time to triage, time to diagnosis and time to essential care as proxies for determining third delay. Time to triage here referred to the time from patient's first contact with a healthcare worker (clinical officer, nurse, medical officer or intern) to the time of departure to the next appropriate care point. Between this time and the time of an objective diagnosis was termed time to diagnosis. The time from when an objective diagnosis was made to the administration of appropriate care or intervention was defined as time to essential care.

### Data analysis

The audio recordings and notes were transcribed verbatim in English then deidentified and imported into NVivo V.12 Plus to generate nodes and to enable thematic analysis[38–40] using inductive coding and a code book was developed. Four of the authors—OOO, TMW, EM and JM—participated in coding transcripts. Separately, at first, we used inductive coding to determine fundamental problems in the care critically ill patients currently. Next, we compared the themes with the Delphi that defined EECC backed with clinical input from the team to identify more EECC-appropriate information. Subsequently, we developed a new code book reflecting both the development of the grounded themes and the EECC themes. Transcripts were then recoded using our new framework.

We then abstracted data relevant to the domains in the third delay. These were statements speaking to equipment, space, infrastructure, administrative processes, staffing, skills, triage and diagnostics. The IDI findings were used to build on the observations during the actual patient journeys. We initially read the patient journeys to provide context and further depth in IDIs. We used the narrative form of the patient journey to further develop and test our analytical framework for the IDIs on the provision of care in critical illness.

### Patient and public involvement

The study design was informed by contemporary experiences of patients in Kenyan public hospitals, especially in light of the COVID-19 pandemic. Throughout the recruitment and conduct of the study, informed consent from patients and/or their relatives was maintained.

We held three feedback meetings with the hospitals after completion of data collection where we shared our findings and checked if our interpretation of the data made sense to hospital staff. Two of these (facility 1 and facility 2) were held in-person at the respective facilities, while one (facility 3) was held virtually via Microsoft Teams. Facility 4 was unable to take part due to their COVID-19 workload. The audiences consisted of healthcare staff from various departments including Accident and Emergency, outpatient, surgery, paediatrics, medical ward, laboratory, radiology, nutrition and hospital administration. The teams confirmed that our findings were a fair representation of their respective hospitals.

### RESULTS
### Themes

The following are findings from the 12 IDIs with the 11 healthcare workers and 4 patient journeys. The combined actual patient journeys are separately attached as online supplemental appendix 4. The main findings in this study were combined into three themes: suboptimal systems for identification of critical illness (subthemes—informal triage systems (time to triage) and delays in early recognition of condition (time to diagnosis)), inadequate resources for initiation and continuity of essential care (subthemes—resources and infrastructure contribution to delay and shortage of a skilled health workforce) and disruption of the flow of care (subthemes—inflexible administrative processes (registration and payments) and inefficient communication systems). These findings are summarised in the theoretical model we developed in this study (figure 3).

Table 2 outlines the steps from a healthcare worker's perspective of the journey of a hypothetical patient presenting to the Accident & Emergency department, typically as they navigate the various care points across the hospital departments of one of the study hospitals. The minimum and maximum time are estimates from the healthcare worker's point of view.

### Suboptimal systems for identification of critical illness
#### Informal triage systems (time to triage)

Overall, all four hospitals had a designated area for triage of patients as they arrive at the casualty (Accident and Emergency) and outpatient departments and processes to appraise a patient's vital signs—temperature, blood

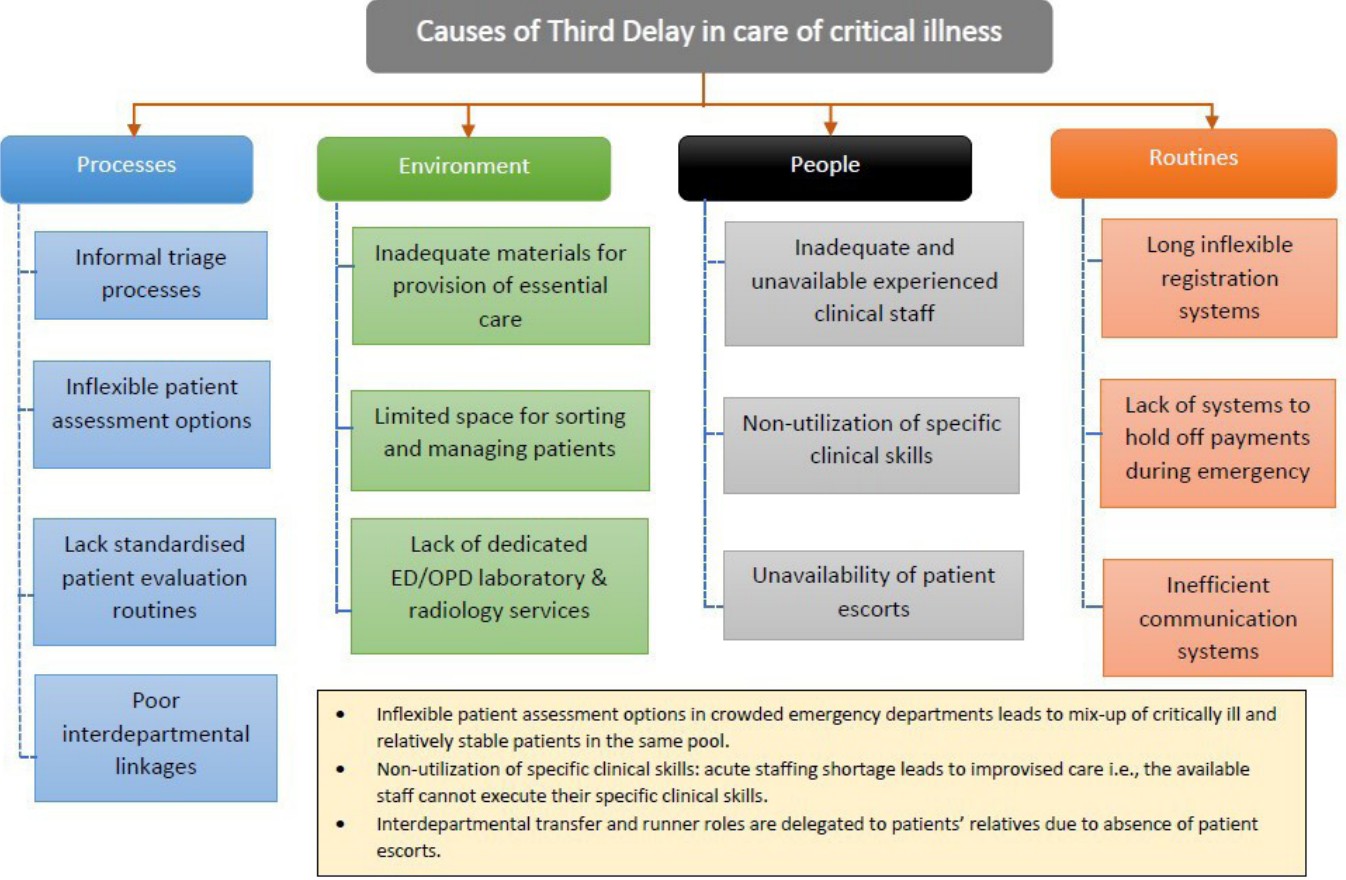

**Figure 3** An empirical concept of the causes of third delay in the care of patients presenting with clinical illness. ED, emergency department; OPD, outpatient department.

pressure, pulse rate and oxygen saturation levels—and general status within twenty minutes of arrival (table 2, boxes 1 and 2) at the emergency departments. However, there was a consistent lack of routines and systems for identification and continuous patient monitoring after the initial evaluation.

> Okay for that one [triage] it's not really clear. It is not clear; there is a no systematic triaging.—*HCW3 (Clinical Officer– Facility 1).*

At facility 1, for example, on entry into the facility, the patients were directed into the tent where they waited to be seen—mostly on a first-come-first-served basis. The tent was located away from the general outpatient and emergency departments and was set up at the advent of COVID-19 to screen patients. The tent was reported to be congested with no specific guidelines governing the categorisation of these patients in the tent.

> […] there is delayed care because […] you know you have to remain in the triage until you're investigated for COVID at times there are no swabs, so you'll remain there. You see, the lab is far so even when they get their samples they have to go to the lab, back again, take care of other patients along the way you find that the patients can stay outside the tent for

even twelve hours so there is delayed care.—*HCW3 (Clinical Officer—Facility 1).*

While the hospitals had allocated physical space—room or cubicle—in the emergency and outpatient departments for triage, time to triage was markedly increased and effective triage practice was mostly postponed or forfeited altogether. Triage was largely informal, and the flow of patients was dependent on the availability and number of the staff and resources, as facilities adopted routines that worked most conveniently for these understaffed departments. The stretched capacities in space and staff-to-patient ratios further compounded the effects of ineffective triage.

> Yes. Then they issue of congestion because we have outpatient, which is there, we have the emergency department, we have the…they are congested in one place…*HCW6 (Clinical Officer—Facility 4).*

> Ok in the triage system, sometimes you find that when we the staffs, there is shortage of staff. Like say you are alone for example like in the paediatric casualty; our acute room is a bit far from where we do triage so when you're handling a patient in the acute room and there is nobody at the triage so you might miss out identification of a critically

**Table 2** A sample patient journey from the perspective of healthcare workers at hospital 1 (this is a hypothetical patient who ends up needing CT scan)

| Step | Activity | Minimum time (min) | Maximum time (min) |
|---|---|---|---|
| 1 | Patient arrives in the casualty department | | |
| 2 | Patient is received by the triage nurse | 1 | 10 |
| 3 | Preliminary vital signs are taken by the nurse | 1 | 2 |
| 4 | Patient waits in the queue at the registration window | 1 | 30 |
| 5 | Confirmation of patient health cover/payment of registration fee | 2 | 5 |
| 6 | Patient registration | 5 | 10 |
| 7 | Waiting in the queue for vital signs checking | 1 | 30 |
| 8 | Vital signs repeated and recorded by nurse | 2 | 5 |
| 9 | History taking by nurse | 5 | 20 |
| 10 | Waiting in the queue for review by clinician | 5 | 45 |
| 11 | History taking by clinician | 5 | 15 |
| 12 | Examination by clinician | 5 | 15 |
| 13 | Investigations (CT scan) ordered | 2 | 2 |
| 14 | Transport outside facility to seek the required investigation | 15 | 60 |
| 15 | Waiting in queue for investigation | 30 | 120 |
| 16 | Waiting for interpretation of results | 15 | 60 |
| 17 | Travel back for review with CT scan results | 15 | 60 |
| 18 | Waiting in queue for review with results | 15 | 45 |
| 19 | Doctor's review with results | 10 | 45 |
| 20 | Admission by nurse: nurse calls receiving ward for bed space | 5 | 30 |
| 21 | Waiting for availability of bed space | 5 | 15 |
| 22 | Nurse transports patient to destination ward | 5 | 25 |

**Consultation protocol**
► Clinical officer (±clinical officer interns) sees patient
► Clinical officer calls medical officer (MO)/registrar
► MO/registrar calls consultant

**Investigations**
► Available tests: Random blood sugar (RBS), full hemogram (FHG), urea, electolytes and creatinine (UECs), liver function tests (LFTs), ultrasound, X-ray, magnetic resonance imaging (MRI) (on booking)
► Not available: CT scan, oesophago-gastro-duodenoscopy (OGD)

**Referral protocol**
► Through respective specialties:
  – Consultant reviews patient, decides referral
  – Unit manager notified
  – MO/registrar calls destination hospital with patient details
  – Ambulance set up and patient referred

ill patient in the queue so such scenario, we might end up losing a patient while still on the queue.—*HCW1 (Nurse In-charge, Paediatric Department—Facility 2).*

With the overwhelming numbers of critically ill patients presenting in the emergency departments in the hospitals, one hospital activated a functional formal triage system, adopting the South African Triage System in the emergency department, using colour-coding—red, orange, yellow and green—to classify patients on arrival.

Initially we didn't have a proper triage system, but we brought it up that we needed a proper triage system and for the past I think three or four days we've been having a proper triaging system, we have like the triage sheet where you see and then we have the list of patients who are emergency patients who patients can wait and like that.—*HCW5 (Medical Officer, A&E department—Facility 3).*

So, we are using the South Africa triage system whereby we have at least started coding the patients. We are using the red, orange, green and yellow. So, when a patient comes, we are using the triage chart, which is at the triage area…so we follow that triage, we fill everything and any patient who appears to be red there is…we tag the patient then we take him or her to the emergency area.—*HCW9 (Nurse In-charge, A&E department—Facility 3).*

| Box 1 Case 1: The journey of an adult female patient presenting with progressive abdominal pains at facility 1 |
|---|

Context:

A county referral hospital. The A&E (ER) unit has five nurses on the shift—although one of them is the unit manager or 'in-charge' and, as such has duties other than front-line care. The unit also has two clinical officers and one medical officer. The ER has five couches, four oxygen ports, and there are two bedside monitors that are operational.

An adult female patient arrives at 10:20 hours, accompanied by two relatives, one of whom is her sister. The nurse takes a brief history and identifies symptoms of severe abdominal pains. Four min after the patients' arrival, the nurse takes vital signs and the patient's oxygen saturation (SPO$_2$) level is noted to be 93% on room air, and she has a slightly elevated heart rate of 90 beats per min.

Twenty min later, the nurse takes her blood pressure measurement, and it is noted to be 93/50 mm Hg. The patient is also found to be clinically pale. The nurse sets up an intravenous infusion of normal saline. At this point, the clinician has ordered a laboratory request for a full haemogram, but the sample is not delivered to the laboratory until the patient's relatives make payments for the laboratory investigations—at 12:04 hours, nearly 2 hours after she arrived.

Over 2 hours after the initial vital signs, the medical officer reviews the patient and orders an abdominopelvic ultrasound. The relative joins the queue at the payment window to make payments for the investigation. The patient, in a wheelchair, then joins the queue at the radiology department.

At 13:45 hours, a staff member at the radiology department sends the patient's relative back to the A&E to communicate that the test cannot be done at the moment because the patient is supposed to have fasted. The back-and-forth between these two departments continues until 14:15 hours when the doctor at the A&E orders specifically for a pelvic ultrasound. This is now 4 hours after the patient arrived.

Afterward, the patient is wheeled back to the radiology department and the relatives queue up at the payment window to update payments for the new test. The patient remains in the queue until 17:33 hours when she is wheeled into the ultrasonography room. A diagnosis of ruptured ectopic pregnancy was made, and the patient was admitted 12 hours later, at 22:30 hours, for an 'urgent' surgical intervention: an exploratory laparotomy.

A&E, Accident and Emergency; ER, emergency room.

| Box 2 Case 2: The journey of an adult male patient brought in unconscious at facility 2 |
|---|

Context:

The unit has ED has eight beds, all occupied at the time the patient is brought in. this department has three nurses, one medical officer and two emergency medical technicians (EMTs). The unit is full of patients: some are seated on chairs, others waiting outside on benches. The department lacks piped oxygen but has three oxygen cylinders with flowmeters attached (no flow splitters). The three cylinders are all in use at the time the patient is brought in.

Four min after arrival, at 12:10 hours, the nurse takes vital observations, and the SPO2 is 85% on room air. At 12:22 hours, the patient is placed on a bed, but he is sharing the bed with another patient since the space is full. The patient awaits review. No intervention since the initial vital observations at 12:10 hours. Ten min later, the medical officer instructs EMTs to fix an IV line and draw blood samples for laboratory investigations.

At 12:35 hours, the medical officer writes a laboratory request and hands the blood samples to the patient's relatives to deliver to the laboratory on making payment for the tests. Twenty min later, a nurse sets up a 500 mL bottle of IV Ringer's Lactate. The patient is yet to be reviewed.

At 13:25 hours, the patient has been noticed to be having convulsions. We still have not had a concrete review because the doctor is quite held up and the patient's details have not been updated. Fifteen min later, the medical officer prescribes an anticonvulsant. The prescription is handed to a relative to pick up the drug from the pharmacy on payment. The unit is still very busy, with very sick patients being wheeled in. The medication arrives 1 hour later, at 14:41 hours and is administered 6 min later. All this while the patient has not yet been reviewed and is still in bed in status quo; still unresponsive and having convulsions. The results from the lab are not yet out.

At 16:15 hours, the results are yet to be received back from the lab. No further vitals have been taken. The patient's condition has not improved much, but the convulsions have subsided for the moment. The patient is yet to be reviewed.

The observer stopped the observation at 16:30 hours as his shift ended. ED, emergency department.

## Delays in early recognition of condition (time to diagnosis)

Although the hypothetical journey map (table 2) placed time to diagnosis within 35 min, findings from our actual patient journeys found that for tests available in the hospitals, it took 3–12 hours to make a clinical diagnosis after triage and initial evaluation. In-between the time the tests were ordered and when the results came back, the management of the patient was mainly symptomatic, with no system for continuous evaluation of patients' conditions (boxes 1–3).

Besides being limited in physical space, the diagnostic departments were noted to be lacking variety of tests, occasioning profound delays and often necessitating patient referrals out of the hospitals to source the services. These can be seen in the hypothetical journey map, cases 1 and 2, and in the following quotes from IDIs.

We only have X-ray, and ultrasound. Yeah, so we don't have a CT scan yet…and MRI we don't have. If we want to do an MRI or a CT scan we have to send outside.—*HCW1 (Nurse In-charge, Paediatric Department—Facility 2).*

Then the other thing is the laboratory. Our laboratory lacks many things; like there is no culture and sensitivity, those are things that miss in our lab. So, if we can have a bigger lab with many areas of assessment, it can be very much okay.—*HCW10 (Nursing Services Manager—Facility 1).*

With most hospitals having a centralised payment point, processing the laboratory and radiological tests ordered took considerable amounts of time. The delays were occasioned by long, inflexible registration and payment processes and long queues in these departments.

…there could be some delays depending with…maybe the patient needs a CT scan, you find the relative

**Box 3    Case 3: the journey of an adult male patient brought in with symptoms of severe difficulty in breathing at facility 4**

Context:

The emergency room (ER) and the general outpatient are housed under one roof. The unit has one nurse and one medical officer (MO) and is full of patients waiting to be seen. The nurse reported at 7:30 hours and will oversee this unit until 12:30 hours when another nurse will come. The MO is also covering theatre in case of any surgical emergencies. The ER has three[3] beds/couches for patients, and at the moment, all three beds are occupied. There are also a few nursing students assisting in the care of these patients. Also, two clinical officers; one here in the ER, the other one covering the outpatient section. The unit has one functional port for piped oxygen which is currently in use. The ER is full. There is a COVID-19 isolation ward but currently that is full as well. There is no departmental phone. An elderly male patient—a referral on follow-up for COVID-19 pneumonia—arrives at facility 4 at 08:57 hours with symptoms of severe breathing difficulty, although the SPO2 levels were above 95% on arrival. The patient is kept in the emergency department, awaiting allocation of space in the COVID-19 isolation ward at the facility. At 12:17 hours, the MO reviews the patient and noted the patient to be desaturating; SPO2 as low as 84%. There is only one oxygen port at the ER, and at the moment it is in use by another patient. The MO calls another facility, facility X (a regional referral facility), since the isolation unit at facility 4 is full. Facility X responds with a promise to call back once a space avails. At 12:52 hours, the SPO2 levels are noted to be below 80% and the patient is put on oxygen via face mask. The space at facility X has not been confirmed yet. There is a back-and-forth between the medical officer and facility X until 15:43 hours when the space in the isolation unit at facility X is confirmed, and the ambulance is set up to ferry the patient.

they don't have the money to do the CT scan or any other investigation which is not done with the hospital, so those are some of the challenges we get.— *HCW1 (Nurse In-charge, Paediatric Department—Facility 2).*

## Inadequate resources for initiation and continuity of essential care

### Resources and infrastructure contribution to delay

Although the vital observations were done within 20 min of patients' arrival at the emergency departments, subsequent monitoring was delayed or not done altogether. The reasons cited for this included limited physical infrastructure and shortage and/or failure of equipment. With the surge of patients needing constant monitoring during the COVID-19 pandemic, this proved to be a major challenge across the four facilities as illustrated in the contextual information of the patient journeys (boxes 1–3) and the following IDI quotes.

Like we have only one monitor and we encounter many patients, and you see you cannot put this one [patient] on a monitor then take it from them to connect to another patient. We have one monitor. The BP machines are also limited; they are like two. The

equipment is there but we cannot say it's enough.— *HCW2 (Nurse—Facility 1).*

The few equipment available was sometimes broken down and often shared between the many patients and occasionally between departments. This further impacted negatively on patient monitoring and care and slowed down interdepartmental or interfacility transfers.

…but sometimes you find the SPO$_2$ machine is not working, sometimes we don't have thermometers so we might miss out the degree of fever, so things like that. Sometimes they break down or they're stolen. Like the glucometer—you might travel with the patient but now if you can't check on the blood sugar level this patient might be in hypo[hypoglycaemic], but you'll fail to note.— *HCW9 (Nurse In-charge, A&E department—Facility 3).*

Lack of essential commodities and consumables in the destination departments resulted in slow movement of patients from the outpatient and emergency departments, further slowing down the identification and management of incoming critically ill patients.

I've seen only one point in Medical Ward. Oxygen is a challenge in the wards; most of the wards. And that is why you find patients of COVID stay here [Emergency Department] longer because they [wards] don't have piped oxygen like we, we have. They have to carry a cylinder. And right now, there is no oxygen.— *HCW9 (Nurse In-charge, A&E department—Facility 3).*

As this was in the heart of the COVID-19 pandemic, in some cases, adaptive measures like referral and out-of-pocket purchase of these commodities were used to override the shortages and delays occasioned by unavailability of space and oxygen.

Like recently, the last three four days we didn't have oxygen. So, you tell the patient to make an option where he or she can go to, she can be treated, either Facility 88 or Facility 99 [private hospitals], because we cannot handle them. In real sense, this is a government institution, whereby we are supposed to handle those COVID patients when they come.— *HCW8 (Nurse, A&E department—Facility 4).*

### Shortage of a skilled health workforce

Across the hospitals, human resource shortage was a major hinderance to implementation of care interventions, and the available few staff could not allocate adequate focus to the critically ill patients in these departments.

The other thing I would require staff who are very well trained. We would require protocols and all staff need to be trained about these protocols. That if a patient presents with this symptom, what next? You know, people don't just walk around without you know, knowing what to do.— *HCW7 (Medical Officer—Facility 4).*

In addition, the facilities lacked an elaborate training system for staff on emergency and critical care, and since most of the staff train on the job, it was difficult to keep track of staff readiness to provide care to the critically ill patients.

> And also, many of us, you see we have not been trained all of us about the BLS and ACLS, those ones. Not all of them have been trained; so, if we can get any update training and motivation, that is, it can be very good. Because once you have knowledge, the skills and the technique, those are the three key things you'll need for production.—*HCW10 (Nursing Services Manager—Facility 1).*

In addition, training and retention of skill-specific workforce was picked as a challenge across the hospitals. Constant staff rotations compounded this skill dilution as the trained staff were moved to other departments without replacement of the skillset.

> They keep on changing, rotations…You were in this department, next month you're taken to another department. So, you find that you can be trained in handling emergencies and…you're taken to another department so whoever is replacing you has not been trained so things like that.—*HCW1 (Nurse In-charge, Paediatric Department—Facility 2).*

Sometimes the strained staffing capacities led to coping measures such as rationing of care even seriously ill patients, in order to spread out the care to as many patients as possible.

> In the wards there is also shortage, so they just check in general, and they if they deteriorate there is resuscitation and they make it, sometimes they don't make it–*HCW2 (Nurse—Facility 1).*

### Disruption of the flow of care
#### Inflexible administrative processes (registration and payments)
The hospitals lacked systems to override payments for tests, investigations and essential services for critically ill patients, leading to delays in receipt of adequate and appropriate treatment by patients. Payment for registration, laboratory investigations and radiological tests were a barrier to quality of care and smooth flow of patients between care points. Prescribed tests and investigations were not processed until the patients, or their relatives paid for these services and in between, care was often withheld while the payments were sorted out.

> The issue that we have is this issue of funds. A patient may come to you presenting with hypoglycaemia and you don't know…it's an altered mental status. You really don't know, because the…the relatives have to pay for the RBS [random blood sugar]. You understand?—*HCW7 (Medical Officer—Facility 4).*

Staff in some facilities felt that it would help if there were systems to hold off payments for basic tests until later when the results are out, and the patient has been attended to.

> I would require that we have a very clear path with the lab so that we don't require patients to pay out for things like RBS, you know. Essential tests like BS [blood smear] for P. falciparum. So that in that clear path patients can always pay for these things later once, once they get well or whether they make it, or they don't make it—*HCW7 (Medical Officer—Facility 4).*

Some facilities reported having a waiver process, through the office of the hospital Social Worker, if a patient is unable to pay for a test or investigation. However, even in the facilities that had waiver systems for patients who cannot afford a service, the processes were often very slow and unreliable in the event of critical illness.

> So, the challenge is money…money, and you see the patient cannot be waived, nowadays, you have to pay cash. And when you go to the social worker, what he'll tell you is that he or she can waive to a certain point—not beyond maybe I don't know, 2000 [Kenyan shillings], not beyond 3000. That is now the challenge.—*HCW9 (Nurse In-charge, A&E department—Facility 3).*

#### Inefficient communication systems
We identified ineffective communication systems as a major limitation to reducing the *time to essential care* of critically ill patients. Lack of direct interdepartmental communication lines was picked across all four hospitals. Most departments lacked a functional departmental telephone, and communication between departments was largely through the health workers' personal mobile phones (box 3).

> In fact, I was forgetting. That is a very big problem. We don't have communication equipment. It used to be there but they all broke down so right now we use our phones […] It's very difficult because you have to call personal phones. Somebody can even ignore.—*HCW2 (Nurse—Facility 1).*

In the four facilities, communication generally relied on the office of the covering nurse even in emergencies. This office serves as a call centre for the hospital to provide liaison between departments and to link the hospital with other facilities for inbound and outgoing referrals. In every shift, there is a designated nurse who takes up the role, with a designated facility phone. The covering nurse receives and relays information between the relevant personnel or departments, including the administration when needed. With the absence of processes to maintain care in-between, the protocol for escalation of care was also described as long and time-consuming, especially during night shifts and weekends.

> The nurse now has to look for the nurse covering and tell the nurse covering that we have a patient,

and this patient is to be reviewed by a medical officer. Then the nurse covering calls the medical officer, at the same time the nurse covering calls the driver… At times it takes one hour, two hours, three hours. And sometimes the medical officer is within the hospital yes but doing an operation. So, it means the patients have to wait until the operation is over…That is what happens in the night. In…over the weekend, again there is that delay.—*HCW8 (Nurse, A&E department—Facility 4).*

When probed about the possibility of a better and more prompt communication system, some of the facilities reported that it would be a welcome upgrade, although budgetary constraints would cripple the implementation and sustainability of such a system.

…Maybe if that one [communication system] can be put in place will be able to help our patients.—*HCW1 (Nurse In-charge, Paediatric Department—Facility 2).*

In some cases, the faulty communication hardware meant that the staff had to walk to other departments to relay the information in person when planning on transferring patients to the wards from the emergency department. This is not only time-consuming but also takes away valuable manpower from the already understaffed units.

And it's…it's also difficult because of shortage every time I keep on walking in the ward, 'I'm bringing a patient! I'm bringing a patient!' You know, it's also cumbersome.—*HCW9 (Nurse In-charge, A&E department—Facility 3).*

In general, time to essential care was between 4 and 12 hours. In case 1, for example, a follow-up call the following day revealed that the patient was admitted past 22:30 hours and prepared for an emergency surgical intervention, although the patient had arrived at the facility more than 12 hours prior.

## DISCUSSION

We qualitatively examined the patient monitoring and evaluation processes and the limiting factors in the journey of critically ill patients to identify delays in the delivery of care when seeking care in Kenyan secondary-level public hospitals. We found that while the hospitals had allocated physical space in the emergency and outpatient departments for triage, time to triage was markedly increased and effective triage practice was mostly postponed or forfeited altogether. Our study also identified inadequate resources, ineffective communication systems and inflexible administrative processes—bureaucratic non-clinical and business-related organisational procedures such as payments and registration processes—as the major limitations to reducing the time to diagnosis and time to essential care to critically ill patients within low-resource hospital settings.

Effective management of critically ill patients, that is, initiation and maintenance of essential care, requires quality care sustained over hours or days including periods spent in emergency departments, during transfers, during waits in diagnostic departments for investigations and sometimes prolonged periods spent on the wards.[41] This quality of continuous care is often reliant on front-line health cadres, especially nurses.[42] However, shortage of skilled healthcare workforce is evidently an ongoing challenge across public hospitals. This is not only with regard to staff-to-patient ratios, but also adequacy of knowledge and skills to implement essential and timely care. This has been attributed to low rates of staff employment and high staff turnovers in addition to discordance in training of front-line staff. The effect of this chronic shortage is development and normalisation of adaptive responses such as rationing of care, as the healthcare workers focus on clearing the overwhelming queues with the available resources in these resource-limited hospital settings.[13 19 21 22]

Initial and subsequent evaluation and review of critically ill patients is limited by the chronic shortage of material resources and equipment such as blood pressure machines, thermometers, pulse-oximeters and also by infrastructural limitations such as physical space and seating for examination of patients on arrival to the study hospitals. These inadequacies not only have an immediate and direct effect on the prompt care of patients—as seen in the lengthy delays in triage, diagnosis and definitive management—but they also have normalised coping mechanisms that further constrain the capacity of the hospitals to improve. All these culminate into a marked third delay for patients seeking care in these hospitals. In the case of critically ill patients, the delays could be detrimental to their clinical outcome and lead to preventable mortality. Measures such as a package of interventions focusing on adequate priority infrastructure and medical supplies, prioritising steps such as triage and stabilisation of critically ill patients, sufficiently trained healthcare providers, evidence-based care and referral capacity to support transfers to higher-level care[9 22] are, therefore, key in addressing third delay in hospitals in resource-limited settings.

Evidence suggests that patient-specific information, particularly in the emergency departments, is an essential component of the patient pathway, the patient–physician relationship, nursing practice and team working.[43] However, the process of obtaining and availing required patient details should not impede the initiation and/or flow of care, especially in critical illness. While it is vital to account for all the services offered to patients in the facilities, there should be straightforward systems in place to ensure that the payments, or lack thereof, do not interfere with the care delivered to critically ill patients and that care is not withheld when the patient is not able to raise the amounts required, or when the payments are yet to be processed. For critically unwell patients, open and functional interdepartmental linkages

and communication systems are vital owing to the time-critical nature of their conditions, and the fact that they often cannot advocate for themselves. If the turn-around-time is to be reduced, it is, therefore, essential to eliminate these administrative stumbling blocks, to reduce the congestion in the emergency care pathways, and capitalise on the cost-effectiveness of the basic emergency care interventions.[44–47]

In chaos, the risks are profound. COVID-19 was pivotal to illuminating areas for improvement in ill-prepared hospital systems globally. In summary, the findings from our study speak to a series of dysfunctional systems for care with adaptive mechanisms that are unideal for the care of critically ill patients. Implementing a guiding principle that the most time-critical, life-saving and feasible care is the priority when resources are scarce might be a helpful starting point in improving quality of care and outcomes in hospitals. Such a policy, as articulated in the recent national strategic plan for implementing EECC in Tanzania,[48] could be useful for the change needed to address defective triage, improper initial and continued evaluation of critically ill patients, strained resource capacities and administrative bottlenecks that all impede the flow of care for critically ill patients while seeking care in these secondary-level public hospitals. We believe that examining critical illness care through a third delay lens would allow us to develop better targeted interventions that improve the likelihood of receiving essential and timely critical illness care services. Figure 3 summarises our concept for the causes of third delay in care of critically ill patients.

### Study limitations and strengths

Our study took place during the COVID-19 pandemic and access was limited due to movement restrictions. The research was, therefore, conducted in only four hospitals and was limited to 12 interviews and 7 actual patient walk-through journeys. A wider sample of facilities, healthcare providers and patients would likely provide more robust evidence to explore our aims and support our conclusions. Again, owing to the pandemic, there may have been significant, temporal differences in hospital operations that were not identified as such by the respondents and this may have influenced the management of critical patients in ways that were not recorded here. We, therefore, advocate for follow-up work. The patient journeys method, as we practised it, is also novel and though very revealing in this study, deserves further refinement. Additionally, although the use of video for interviewing during COVID has been justified,[49] it is suboptimal relative to in-person interviews.

However, this study draws strength from the multiple methods employed, which with triangulation, yielded valuable evidence and support. In addition, the study was conducted in facilities operating at the same level from four counties across different regions of the country, which further fortifies the evidence. Lastly, the agreement of various staff from the hospitals—including administrative staff and unit managers—with our findings as being a true reflection of their various facilities provided a validation for our study findings.

### CONCLUSION

Public hospitals in Kenya have long-standing ineffective intrafacility and interfacility systems that are unreliable in the care of critically ill patients when they present in hospitals. Care of critically ill patients—identification, monitoring and decision-making—requires prompt systems to identify and communicate the urgency across multiple care points. Eliminating third delay using a sustained package of timely essential care is, therefore, an important time-critical and cost-effective measure to improve the care outcomes of critically ill patients in resource-constrained settings. Findings from our study can help emphasise the need for standardised effective and reliable care priorities to maintain of care of critically ill patients within hospitals.

**Author affiliations**
[1]KEMRI-Wellcome Trust Research Programme Nairobi, Nairobi, Kenya
[2]Nuffield Department of Medicine, University of Oxford, Oxford, UK
[3]Department of Global Public Health, Karolinska Institutet, Stockholm, Sweden
[4]Centre for Clinical Research Sörmland, Uppsala University, Uppsala, Sweden
[5]Department of Medicine, Nyköping Hospital, Nyköping, Sweden
[6]Ifakara Health Institute, Dar es Salaam, Tanzania, United Republic of
[7]Department of Clinical Research, London School of Hygiene and Tropical Medicine, London, UK
[8]Department of Emergency Medicine, Muhimbili University of Health and Allied Sciences, Muhimbili, Tanzania, United Republic Of
[9]Department of Anaesthesia and Critical Care, Muhimbili University of Health and Allied Sciences, Dar es Salaam, Tanzania, United Republic of

**Acknowledgements** We are grateful to Professor Claudia Hanson – Senior Lecturer, Karolinska Institutet, Sweden; Professor, London School of Hygiene and Tropical Medicine, UK and Dr Roy Nobhojit – Surgeon and Researcher, BARC Hospital, HBNI University, Mumbai, India, for their invaluable mentorship that helped shape the concept and direction of this study. We wish to thank the POETIC-COVID team and the entire EECC network for the immense support and teamwork throughout the study period and beyond. We would also like to acknowledge the contribution of the Ministry of Health – Kenya, the participating hospitals and their staff in the process of this study.

**Contributors** OOO conceptualised and designed the study, acquired and analysed the data, and developed the first draft of the manuscript with assistance from TMW, JNO and JMcK. TB, ME and COS contributed to the conceptualisation and design of the study. TMW, EM, JM, KK and OOO analysed the data together. JNO also contributed to acquisition of data and was the Scientific Lead for the larger POETIC-COVID study. All the authors interpreted the findings, critically revised the manuscript and approved the final version. OOO is responsible for the overall content as the guarantor.

**Funding** This study was funded through the POETIC-COVID project by a grant from Wellcome Trust (221571/Z/20/Z) as part of the Innovation in low-income and middle-income countries flagship. ME is supported by a Wellcome Senior Fellowship (#207522) with further support for authors from a core grant awarded to the KEMRI-Wellcome Trust Research Programme (#092654).

**Competing interests** None declared.

**Patient and public involvement** Patients and/or the public were involved in the design, or conduct, or reporting, or dissemination plans of this research. Refer to the Methods section for further details.

**Patient consent for publication** Consent obtained directly from patient(s).

**Ethics approval** This study involves human participants and this study was part of a larger study approved by the KEMRI Scientific and Ethical Review Committee (SERU Number 4085) and London School of Hygiene and Tropical Medicine (REF 22 866). Institutional approvals were also obtained at hospital level and/or their relevant County liaison for all four hospitals. Participants gave informed consent to participate in the study before taking part.

**Provenance and peer review** Not commissioned; externally peer reviewed.

**Data availability statement** Data are available on reasonable request. Additional anonymised transcripts of patient journeys, process maps and in-depth interviews are attached separately as online supplemental material submitted alongside this manuscript. The raw data supporting this article will be availed by the authors whenever necessary.

**Open access** This is an open access article distributed in accordance with the Creative Commons Attribution 4.0 Unported (CC BY 4.0) license, which permits others to copy, redistribute, remix, transform and build upon this work for any purpose, provided the original work is properly cited, a link to the licence is given, and indication of whether changes were made. See: https://creativecommons.org/licenses/by/4.0/.

**Author note** This is my first ever publication as a lead author and I am immensely proud that it was on the BMJ Open – OOO.

**ORCID iDs**
Onesmus O Onyango http://orcid.org/0000-0003-4045-857X
Carl Otto Schell http://orcid.org/0000-0002-7904-1336
Tim Baker http://orcid.org/0000-0001-8727-7018
Mike English http://orcid.org/0000-0002-7427-0826
Jacquie N Oliwa http://orcid.org/0000-0002-4575-2447

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
