## [Reviewer comments · BMJ Open]

ARTICLE DETAILS

TITLE (PROVISIONAL)	Third delay in care of critically ill patients: a qualitative investigation of public hospitals in Kenya.
AUTHORS	Onyango, Onesmus; Willows, Tamara; McKnight, Jacob; Schell, Carl Otto; Baker, Tim; Mkumbo, Elibariki; Maiba, John; Khalid, Karima; English, Mike; Oliwa, Jacqueline

VERSION 1 – REVIEW

REVIEWER	WONG, Wai-Tat The Chinese University of Hong Kong, Department of Anaesthesia and Intensive Care
REVIEW RETURNED	30-May-2023

GENERAL COMMENTS	- While the third delay is the main concept adopted in the design of this study, the justification to adopt this concept from obstetrics patients to general critically ill patients is unclear. I believed it can be justified with more explanation in the introduction session.- The mechanism to produce the minimum time and maximum time in table 1 is not available (e.g. How many healthcare workers gave opinion?, the way to gather the opinion)- The source of data for table 2 is also unclear.- The total number of nodes and the themes generated from the analysis- It is better to separate the patient journeys and the thematic analysis in the result sessions.- The description in the patient journeys is disorganized and the format is not the same as the patient journeys in the appendixes.- Limitation of the study should be explicitly described (small number of patient journey and limited number of healthcare workers interviewed.- The conclusion is not well corresponding to the main finding of the study. Overall comment, - The result of this study is valuable. A better description of the result can make it a good manuscript.
---

REVIEWER	Lin, Yanxia Shanghai University of Tradition Chinese Medicine, School of Nursing
REVIEW RETURNED	18-Jul-2023

GENERAL COMMENTS	Thank you for inviting me to review this paper. The paper reports a meaningful study trying to understand the delays of care for critically ill patients from admission to diagnosis in multiple settings
---

in Kenya in the context of COVID-19. To strength the report, I suggest the authors to address the following comments:

Title: the title did not specify the method or design of the study. It is not only the journey but also the factors and processes of care related to “third delay”.

Abstract: Again there is no information about the “design”.

Methods did not report the sample size. The usefulness of a package cannot be concluded based on the findings.

Background: There is a lack of contextual information about the country (healthcare system related to critical care) where the study was conducted, with the COVID-19 pandemic. I saw simple description in Study setting section but I think we need evidence as background.

Methods: The design or methodology of the study is unclear. A methodology with a cited reference (or more) is needed to organise all the parts, including data collection, data analysis and findings report.

Study setting: Information about the number of healthcare staff and patients (those critically ill, and infected COVID-19) is needed.

Participants: Inclusion and exclusion criteria for patients and HCPs are needed. Are there any criteria that was used to determine who was critically ill and met the inclusion criteria?

Data collection:

1. It is less clear how “patient journeys mapping” was conducted. Observation? Conversations with HCPs? Is there any pre-defined structure for this mapping process (i.e. the sample journey structure in Table 1)? When and where did the researcher make the audio commentary?
2. Interviews: The semi-structured interview guide needs to be provided as an appendix. Was the interview audio recorded? Was note written for each interview?

Data analysis: The detailed process of data analysis is not provided so a methodology is needed for the reader who then can understand how the themes were developed. Who coded data? How were codes grouped into themes?

Ethics: This information is missing. Ethical approval is needed. How was informed consent obtained?

Results: There were two parts: patient journeys as context, and themes. Two subthemes and six subthemes were identified. I think it is better to place the journeys (text boxes) with related themes which means the journeys to be coded as part of the themes. I can see the authors referred to the journeys in themes, suggesting that they could be a source of codes underpinning the themes. There is also a table with numerous data under a theme (Table 2). So I think all the data can be coded into the themes and then the findings were themes incorporating different sources/ types of data.

The framework developed from the study in Figure 3 is good but definitions for each concept are needed. I think the “systems” listed as the fourth variable need to be reconsidered. In my view, the entire framework of the four concepts is a system. The “systems” used by the authors or said by the participants is in a narrow scope, and if it was used directly in the final result will bring confusion to reader in a common sense.

Theme 1:
 There are two “systems” in the findings which I think are different. The first one is narrow while the other one is broad.
 Subtheme 2 is smaller in size with only two paragraphs and two quotes, compared to subtheme 1.

Theme 2:

	Please maintain the consistency of naming theme across the document. I am not surprised by the findings here about the shortage of staff and resources. Are there any new insights into this challenge? The shortage of resources existed before COVID-19 pandemic, and a barrier to care. What is new in this study? What findings are related to COVID-19 pandemic? Theme 3: Subtheme 1: No findings related to the care itself provided to the critically ill patients. The quote Line 3-8 on page 15 indicates not only shortage of staff but also lack of effective interventions or protocols. So I think here the explanation of data is not complete or systematic. Subtheme 2: Quotes are much longer than authors' commentary narrative in the last subtheme on page 16, which is not recommended in a qualitative study. The content in the last subtheme is very descriptive with limited explanations and interpretations. Discussions: Some content remain 'findings' rather than "discussions". I think it is better to relate the findings back to third delay and or other theories. Explain that whether the third delay exists in critical care in the context of COVID-19 pandemic in this particular country. The implications of the findings to other countries or areas need to be addressed. As the author discussed in Line 6-9 and Line 37-39 on Page 18, to a great extent the findings in this study are same with the existing evidence. So what new knowledge were produced by this study? I think the contribution to knowledge needs to be clearer. References: Some references about factual literature are old.
--	---

VERSION 1 – AUTHOR RESPONSE

Reviewer 1: Dr. Wai-Tat WONG, The Chinese University of Hong Kong

- 1. While the third delay is the main concept adopted in the design of this study, the justification to adopt this concept from obstetrics patients to general critically ill patients is unclear. I believed it can be justified with more explanation in the introduction session.***

Very useful insight. We have added a justification and further description of this in the last paragraph of the Background. Also notable is that we have rearranged the section for better flow. This includes further information of Kenyan context around the time of the study.

- 2. The mechanism to produce the minimum time and maximum time in table 1 is not available (e.g. How many healthcare workers gave opinion? the way to gather the opinion)***

Thank you. We added a more detailed description of the process mapping and patient journeys on page 6 in the methods section. This is captured in page 7, line 16-23.

- 3. The source of data for table 2 is also unclear.***

These data were obtained during health facility assessments (HFAs) which were the preliminary stage of the larger project within which this study was conceived. This information has been included *in the methods section*, page 7, line 16-17.

4. *The total number of nodes and the themes generated from the analysis*

I agree. The initial submission lacked clarity on these. We have provided further description of our coding process under Data analysis on page 7, line 42-50.

5. *It is better to separate the patient journeys and the thematic analysis in the result sessions.*

This, I admit, was because we were trying to avoid making the paper unnecessarily long. In the same breadth, we condensed the patient journeys into the format in the textboxes for the same reason (answer to 6 below). We have added these same patient journeys in the appendix to match the others. The second reviewer, however, felt that the textboxes were better of coded together under the themes. We embraced this angle but provided more supporting information to back and link them to the IDI findings.

6. *The description in the patient journeys is disorganized and the format is not the same as the patient journeys in the appendixes.*

As stated in 5 above.

7. *Limitation of the study should be explicitly described (small number of patient journey and limited number of healthcare workers interviewed.*

I think you might have missed this as it was captured and likely hidden under the discussion section. With a few edits, we have moved it up just below the abstract under the Article summary – on page 2, line 40-48.

8. *The conclusion is not well corresponding to the main finding of the study.*

We have rewritten this section for clarity and linkage to the findings. We hope this current version resonates better with the subject and findings.

Reviewer 2: Dr. Yanxia Lin, Shanghai University of Tradition Chinese Medicine

9. **Title:** *the title did not specify the method or design of the study. It is not only the journey but also the factors and processes of care related to “third delay”.*

Thank you, Dr Lin for pointing this out. We modified the title to "Third delay in care of critically ill patients: a qualitative investigation of public hospitals in Kenya".

10. **Abstract:** *Again there is no information about the “design”. Methods did not report the sample size. The usefulness of a package cannot be concluded based on the findings.*

We have added the study design and number of IDIs & patient journeys in the abstract. Page 2, line 8-10.

11. **Background:** *There is a lack of contextual information about the country (healthcare system related to critical care) where the study was conducted, with the COVID-19 pandemic. I saw simple description in Study setting section but I think we need evidence as background.*

Many thanks. We made adjustments – rearranged the section for better flow and also added some contextual evidence. This is found on page 4, line 14-21.

12. **Methods:** *The design or methodology of the study is unclear. A methodology with a cited reference (or more) is needed to organise all the parts, including data collection, data analysis and findings report.*

I agree, the initial submission was scanty in this regard. We have added the study design and related information under the subheading “Study design” on page 6, line 4-9.

13. **Study setting:** *Information about the number of healthcare staff and patients (those critically ill, and infected COVID-19) is needed.*

We have described the study setting in more detail on page 6, line 12-20 and additional information on the staffing structure and number of patients per department in Table 1 in the results section.

14. **Participants:** *Inclusion and exclusion criteria for patients and HCPs are needed. Are there any criteria that was used to determine who was critically ill and met the inclusion criteria?*

We have added this information under the section titled “Participant selection and recruitment” on page 6, line 23-40.

15. Data collection:

1. *It is less clear how “patient journeys mapping” was conducted. Observation? Conversations with CPs? Is there any pre-defined structure for this mapping process (i.e. the sample journey structure in Table 1)? When and where did the researcher make the audio commentary?*

We have separated and described the patient journeys and the IDIs. The patient journeys are described in detail on page 7, line 5-39, under the sub-heading “Process mapping and patient journeys”. The audio commentaries were made onsite by OO, the lead author, at his discretion, using a Dictaphone.

2. *Interviews: The semi-structured interview guide needs to be provided as an appendix. Was the interview audio recorded? Was note written for each interview?*

We have better described the information pertaining to the interviews on page 6, line 43-48 and page 7, line 1-2. The interviews were audio recorded. Notes weren't taken as part of the interview process. The interview guides have been attached as Appendix 2.

16. Data analysis: *The detailed process of data analysis is not provided so a methodology is needed for the reader who then can understand how the themes were developed. Who coded data? How were codes grouped into themes?*

We have provided this information on page 7, line 42-50 and page 8, line 1-9. Four coders, first separately through line-by-line coding then axial coding.

17. Ethics: *This information is missing. Ethical approval is needed. How was informed consent obtained?*

This information was in the initial submission, might have skipped your attention. We have moved it up to page 20, line 33-37. Further information on consent and the process is explained on page 6, line 29-40 under "Participant selection and recruitment".

18. Results: *There were two parts: patient journeys as context, and themes. Two subthemes and six subthemes were identified.*

I think it is better to place the journeys (text boxes) with related themes which means the journeys to be coded as part of the themes. I can see the authors referred to the journeys in themes, suggesting that they could be a source of codes underpinning the themes. There is also a table with numerous data under a theme (Table 2). So I think all the data can be coded into the themes and then the findings were themes incorporating different sources/ types of data. The framework developed from the study in Figure 3 is good but definitions for each concept are needed. I think the "systems" listed as the fourth variable need to be reconsidered. In my view, the entire framework of the four concepts is a system. The "systems" used by the authors or said by the participants is in a narrow scope, and if it was used directly in the final result will bring confusion to reader in a common sense.

Many thanks for this suggestion. I admit, it was a challenge agreeing how to best present these two in a fashionable and clear manner. We have moved the textboxes under the relevant themes and sub-themes and provided the supporting explanations. In general, we have made some rearrangement in the results as well.

Theme 1:

There are two "systems" in the findings which I think are different. The first one is narrow while the other one is broad.

Subtheme 2 is smaller in size with only two paragraphs and two quotes, compared to subtheme 1.

We have rearranged to include the patient journeys and added some interpretation. We have also added further padding with supporting explanations.

Theme 2:

Please maintain the consistency of naming theme across the document.

(This has been addressed accordingly)

I am not surprised by the findings here about the shortage of staff and resources. Are there any new insights into this challenge? The shortage of resources existed before COVID-19 pandemic, and a barrier to care. What is new in this study? What findings are related to COVID-19 pandemic?

We modified the title to shift the focus from COVID-19 per se. That said, we rearranged the results under this theme and included the patient journey as well. This can be seen from page 12 line 14 to page 14 line 35.

Theme 3:

Subtheme 1: No findings related to the care itself provided to the critically ill patients. The quote Line 3-8 on page 15 indicates not only shortage of staff but also lack of effective interventions or protocols. So I think here the explanation of data is not complete or systematic.

We did some rewriting and included a patient journey. This is addressed on page 15, line 1-27.

Subtheme 2: Quotes are much longer than authors' commentary narrative in the last subtheme on page 16, which is not recommended in a qualitative study. The content in the last subtheme is very descriptive with limited explanations and interpretations.

We have readdressed this to include more explanation and interpretations. We have also reduced the size of some quotes.

19. Discussions:

Some content remain 'findings' rather than "discussions".

I think it is better to relate the findings back to third delay and or other theories. Explain that whether the third delay exists in critical care in the context of COVID-19 pandemic in this particular country. The implications of the findings to other countries or areas need to be addressed. As the author discussed in Line 6-9 and Line 37-39 on Page 18, to a great extent the findings in this study are same with the existing evidence. So what new knowledge were produced by this study? I think the contribution to knowledge needs to be clearer.

Thanks for this. We removed some bits that sounded more of results and summarized the main findings in paragraph 1. We have attempted to reorganize the flow and provided further interpretation and discussion of these findings in subsequent paragraphs.

20. References: Some references about factual literature are old.

Notably, we agree that some of our references are a bit too far back. However, we also appreciate the evolution or constancy of the information reference and had meant to outline the breadth of factual evidence over time. We therefore removed a few that seemed to have been overwritten by newer evidence but maintained some like the reference to the Three Delays, which trace as far back as the '90s. We greatly appreciate this observation and suggestion, however.

VERSION 2 – REVIEW

REVIEWER	WONG, Wai-Tat The Chinese University of Hong Kong, Department of Anaesthesia and Intensive Care
REVIEW RETURNED	24-Oct-2023

GENERAL COMMENTS	Methodology session: Please elaborate the reasons of not including the 5 IDI in the analysis (12 out of 17 IDI were used). Result: There are 11 frontline healthcare workers included in the analysis. The labelling of healthcare workers (HCW1, HCW 2...HCW 11) can help to clarify the data. The current inclusion of their role and the facilities number is helpful and should be kept. Please also indicate clearly the number of HCWs and patient journeys involved in the analysis as in the abstract. Discussion: In the second paragraph, the discussion on human resource and other resources (material resources) should be discussed separately, with reference to the findings of the study. In the third paragraph, the connection between the concepts/terms of inter-disciplinary linkage, functioning linkage and communication is unclear. Box 1-3 (case 1-3): There is no indication of the difference between those phrases bolded and highlighted in different colour. Appendix 5: There is no indication of the meaning of the highlighted time.
--

REVIEWER	Lin, Yanxia Shanghai University of Tradition Chinese Medicine, School of Nursing
REVIEW RETURNED	13-Sep-2023

GENERAL COMMENTS	Thank you for inviting me to review this manuscript again. I am happy to see the revisions made by the authors that have improved the paper a lot. I still have the following comments for the authors to consider: Results:  1. Themes: In the first paragraph on page 9 it is better to report the themes using consistent names with the following headings and show the structure of the themes and subthemes. 2. Make sure to name the three boxes in a consistent way, for example, "The journey of a XX patient with XXX". Some texts are bolded or highlighted in red colour. I think use bold only is clearer. 3. Subtheme: "The role of administrative processes in delay-registration and payments". I think this is not a theme but a category. The actual pattern needs to be specified in the name: what role? 4. I recommend to remove Table 2 in Results. They are not the data collected from interviews or observations. The numbers about staff-to-patient ratios can be reported in Study setting section. 5. The themes show a mixed understanding of the Third Delay, with some refers to the consequences due to the lack of resources (Theme 1 and its subthemes), and some are the lack of resources themselves (Theme 2 and its resources). I think the study needs to make contributions (the mechanisms of poor quality of care due to the lack of resources, i.e. barriers and facilitators, and pathways etc.) to the existing understanding about the topic (lack of resources, shortage of staff etc.). Discussion  6. Line 8 Paragraph 1 in Discussion: "...business- and protocol-related organizational processes-as the major limitations to reducing...".
---

	I am not sure I understand here “protocol-related organizational processes”. If they mean interventions or implementations of care, I do not think the study findings support this conclusion. A source of delay might be the care provision or intervention implementation that can be identified from observation or document analysis (scanning the documents or protocols available). This is missing in the study.
--	---

VERSION 2 – AUTHOR RESPONSE

Response to the reviewers: Review 2

Once again, on behalf of all the authors herein, I am grateful for your rigorous and invaluable insights and reviews. We don't take it for granted the time and energy you have vested in ensuring this manuscript is to the best standards to merit publication. We have worked through and addressed the reviews and suggestions and compiled this latest version “**Third Delay_Revision_2**”, and hereby resubmit it for your further appraisal. Alongside the submission is the SRQR checklist and revised Appendices.

Below are our actions we took to address the suggestions shared by each reviewer.

Reviewer 1: Dr. Wai-Tat WONG, The Chinese University of Hong Kong

1. *Methodology session: Please elaborate the reasons of not including the 5 IDI in the analysis (12 out of 17 IDI were used).*

Thank you for pointing this out. We have since realised that it will be confusing to have this detail in this manuscript altogether, since this study speaks to only 4 hospitals. The additional 5 interviews were from a tertiary hospital that was not included in this study. This information has been removed and reframed.

2. *Result: There are 11 frontline healthcare workers included in the analysis. The labelling of healthcare workers (HCW1, HCW 2...HCW 11) can help to clarify the data. The current inclusion of their role and the facilities number is helpful and should be kept. Please also indicate clearly the number of HCWs and patient journeys involved in the analysis as in the abstract.*

This has been addressed. We have labelled the healthcare workers as HCW1 to HCW11. This is captured in the Results section from page 10 to page 16.

3. *Discussion: In the second paragraph, the discussion on human resource and other resources (material resources) should be discussed separately, with reference to the findings of the study. In the third paragraph, the connection between the concepts/terms of inter-disciplinary linkage, functioning linkage and communication is unclear. Box 1-3 (case 1-3): There is no indication of the difference between those phrases bolded and highlighted in different colour.*

This human resources and material resources have been discussed separately on page 18, paragraphs 2 (line 12-22) and 3 (line 24-36). The concepts have been clarified in paragraph 4 (line 38-50) of page 18, which has since been rewritten. The colour highlights have been removed. Only bold used for relevant emphasis.

4. *Appendix 5: There is no indication of the meaning of the highlighted time.*

I agree. This has been duly revised as suggested. The colour highlights have been removed from the patient journeys.

Reviewer 2: Dr. Yanxia Lin, Shanghai University of Tradition Chinese Medicine

5. *Themes: In the first paragraph on page 9 it is better to report the themes using consistent names with the following headings and show the structure of the themes and subthemes.*

We modified these on page 9 line 2-11 to read these main themes suboptimal systems for identification of critical illness (subthemes – informal triage systems (*time to triage*) and delays in early recognition of condition (*time to diagnosis*)), inadequate resources for initiation and continuity of essential care (subthemes – resources and infrastructure contribution to delay and shortage of a skilled health workforce), and disruption of the flow of care (subthemes – inflexible administrative processes (registration and payments) and inefficient communication systems). Thank you, Dr. Lin, for pointing this out.

6. *Make sure to name the three boxes in a consistent way, for example, “The journey of a XX patient with XXX”. Some texts are bolded or highlighted in red colour. I think use bold only is clearer.*

This has been appropriately corrected as suggested on pages 10, 12, 15. The colour highlights have been removed.

7. *Subtheme: “The role of administrative processes in delay-registration and payments”. I think this is not a theme but a category. The actual pattern needs to be specified in the name: what role?*

This has been changed to read “inflexible administrative processes (registration and payments)” on page 15 line 12.

8. *I recommend to remove Table 2 in Results. They are not the data collected from interviews or observations. The numbers about staff-to-patient ratios can be reported in Study setting section.*

Thank you. This has been moved as suggested. Page 6 from line 21.

9. *The themes show a mixed understanding of the Third Delay, with some refers to the consequences due to the lack of resources (Theme 1 and its subthemes), and some are the lack of resources themselves (Theme 2 and its resources). I think the study needs to make contributions (the mechanisms of poor quality of care due to the lack of resources, i.e. barriers and facilitators, and pathways etc.) to the existing understanding about the topic (lack of resources, shortage of staff etc.).*

We have attempted to restructure and further discuss these section as further depicted in our theoretical model.

10. *Discussion Line 8 Paragraph 1 in Discussion: “...business- and protocol-related organizational processes as the major limitations to reducing...”. I am not sure I understand here “protocol-related organizational processes”. If they mean interventions or implementations of care, I do not think the study findings support this conclusion. A source of delay might be the care provision or intervention implementation that can be identified from observation or document analysis (scanning the documents or protocols available). This is missing in the study.*

This has been rewritten for clarity. The intention here was to mean the procedures and processes that are not care-related but hinder the flow of patients. We have rewritten this on Page 18 lines 7-9 to read “...inflexible administrative processes – bureaucratic non-clinical and business-related organizational procedures such as payments and registration processes.”

VERSION 3 – REVIEW

REVIEWER	Lin, Yanxia Shanghai University of Tradition Chinese Medicine, School of Nursing
REVIEW RETURNED	07-Dec-2023
GENERAL COMMENTS	I am very happy to see the revisions made by the authors that have strengthened the paper a lot. Now it is a piece of work in quality that is publishable. As many changes have been made, a final comment is that please read through the manuscript closely for grammar, spelling and formatting accuracy.

VERSION 3 – AUTHOR RESPONSE